# Parenting under the triple burden of violence, depression, and poor diet quality: An intergenerational mother–child syndemic in Nepal

Luissa Vahedi[1,2,3]*, Ilana Seff[3,4], Alexander C. Tsai[5,6,7], Lora Iannotti[3,4], Lindsay Stark[3,4]

**1** The Centre for Global Child Health, SickKids Hospital, Toronto, Ontario, Canada, **2** Factor-Inwentash Faculty of Social Work, University of Toronto, Toronto, Ontario, Canada, **3** Brown School, Washington University in St. Louis, St. Louis, Missouri, United States of America, **4** School of Public Health, Washington University in St. Louis, St. Louis, Missouri, United States of America, **5** Center for Global Health and Mongan Institute, Massachusetts General Hospital, Boston, Massachusetts, United States of America, **6** Harvard Medical School, Harvard University, Boston, Massachusetts, United States of America, **7** Department of Epidemiology, Harvard T.H. Chan School of Public Health, Harvard University, Boston, Massachusetts, United States of America

* l.vahedi@wustl.edu

## Abstract

Parenting while experiencing intimate partner violence (IPV), depression, and poor diet quality can contribute to child health inequities. Syndemic theory offers a framework to understand how maternal conditions can lead to inequitable child health outcomes, such as diarrheal disease: a risk factor for childhood mortality and stunting. This study uses the 2022 Nepal Demographic and Health Survey to test an intergenerational syndemic model addressing three key questions: (1) Do IPV, depression, and absent dietary iron intake co-occur among mothers? (2) Does the probability of child diarrhea increase with concurrent maternal exposures to these three factors? (3) Are these relationships stronger in disadvantaged households? The analysis included 2,019 mother-child dyads. Interaction models on additive and multiplicative scales were constructed to predict child diarrhea as a function of maternal syndemic exposures, accounting for complex survey design. The prevalence of single exposure to IPV, depression, and absent dietary iron intake was higher among mothers in poorer households. In disadvantaged households (two lowest household wealth quintiles), children's odds of diarrhea were higher when mothers experienced syndemic vulnerability (IPV, depression, and absent iron intake) on multiplicative (aOR=1.321, 95%CI:1.000, 1.745) and additive scales (B=0.027, 95%CI:-0.002, 0.055). Household disadvantage may exacerbate the synergistic effects of IPV, depression, and iron deficiency, increasing the likelihood of child diarrhea. Interventions targeting these maternal health adversities in disadvantaged households may reduce the impact on child health.

**Data availability statement:** Data are publicly available from the DHS Program: https://dhsprogram.com/data/dataset_admin/login_main.cfm?CFID=89546425&CFTOKEN=6ecbd-b7283cb26ce-1253329B-9405-770E-A9B4FE-8C5E2166F3.

**Funding:** This work was supported by the U.S. National Institutes of Health (K24DA061696-01 to ACT), the The Social Sciences and Humanities Research Council of Canada (Doctoral Fellowship to LV), and the Philanthropic Educational Organization (P.E.O) (Scholars Award to LV). The funders had no role in study design, data collection and analysis, decision to publish, or preparation of the manuscript.

**Competing interests:** I have read the journal's policy and the authors of this manuscript have the following competing interests: ACT reports receiving financial honoraria from Elsevier Inc. (for his work as Co-Editor in Chief of the Elsevier-owned journal SSM - Mental Health) and from BMJ Publishing Group Ltd. (for his work as Clinical Editorial Advisor of The BMJ). All other authors declare no competing interests.

## Introduction

Diarrheal disease is a major driver of child mortality in low- and middle-income countries (LMICs). The maternal health and caregiving context strongly influences a child's vulnerability to diarrheal disease and symptoms. In particular, intimate partner violence (IPV), depression, and inadequate nutrition are three maternal adversities that may jointly increase the risk of child diarrheal disease. In LMICs, diarrhea is a leading cause of child mortality and morbidity, accounting for about 9% of under-5 mortality [1,2]. Chronic diarrheal disease among children under 5 contributes to acute and chronic malnutrition, which negatively affects the trajectory of development [3,4]. The parenting and household environment can determine to what degree children access health services [5–7]. Among heterosexual couples, mothers bear additional parenting and household responsibilities, compared to fathers, [8] and are also more likely to experience mental health conditions such as depression, intimate partner violence (IPV), and reduced access and intake of nutritious foods [9–14]. Investigating under what conditions women bear parenting responsibilities can elucidate both biological and social mechanisms which explain persisting disparities in the patterning of child diarrhea.

### Interconnections between Maternal IPV, Depression, Poor Diet Quality, and Childhood Diarrheal Disease

Maternal IPV exposure, depression, and nutrition are important risk factors for inadequate child growth and development. Mechanisms between such harmful maternal exposures and child diarrheal disease have not been robustly established in empirical literature. We propose that pathways linking maternal IPV, depression, and inadequate dietary iron intake can jointly shape children's risk of diarrheal disease through parental stress and strained caregiving, above and beyond water and sanitation infrastructure [15–17]. IPV can expose mothers to chronic stress that elevates depressive symptoms and undermines emotional regulation, impairing attachment and reducing the consistency and responsiveness of caregiving [18–22]. Depression is associated with lower energy, impaired attention, and difficulties in maintaining daily routines essential for child hygiene and feeding practices, all of which influence exposure to enteric pathogens that can cause diarrhea. Inadequate dietary iron intake, which worsened in situations of IPV and gender inequitable household resource allocation [23–26], further compounds these effects. Iron deficiency increases fatigue and cognitive strain, which can diminish a mother's physical capacity to engage in energy-demanding caregiving behaviors, including preparing safe foods, monitoring child illness, and breastfeeding [23,27]. We hypothesize that these interactions related to stress and parenting are magnified in more disadvantaged households, above and beyond the presence of clean water and adequate sanitation infrastructure. Within this proposed syndemic, stressful parenting acts as a pathway heightening physiological and psychological burdens, weakening maternal parenting capacity, and increasing the likelihood that children experience unsafe feeding, sanitation, and care practices. These interacting maternal vulnerabilities create a compounded risk environment in which children are more susceptible to diarrheal disease due to reduced protection.

To date, research has investigated maternal IPV, depression, and poor diet quality in isolation of each other and relative to the outcomes of child diarrhea, development, and parenting. [28–30] However, IPV, depression, and nutrition are inter-related phenomena. Parenting under the triple burden of IPV, depression, and poor diet quality may be particularly challenging in LMIC contexts, leading to disparities in child diarrhea that threaten survival. However, maternal IPV, depression, inadequate nutrition and child diarrhea have not been incorporated under a unifying theoretical framework.

### Theoretical framework

The term *syndemic* combines two words: synergy and epidemic. Syndemics occur when two or more co-occurring/clustering epidemics/health conditions interact with one another to exacerbate the population health disparities [31]. Inequitable sociopolitical conditions enable the clustering and interaction of the epidemics/health conditions, ultimately giving rise to the magnified population-level harms [31]. Thus, syndemic theory is encompassed by the three theoretical postulates of clustering, interaction, and the driving role of contextual factors.

The first formally studied syndemic involved the interplay between social, economic, and biological factors contributing to the clustering and mutually reinforcing effects of substance abuse, violence, and HIV (known as the SAVA syndemic) in the Puerto Rican community in Connecticut, USA [32]. The SAVA syndemic is the most widely investigated syndemic globally [33–36]. Accordingly, the literature base in LMICs is dominated by outcomes related to HIV/AIDS: positive test results, viral suppression, and risky sexual practices like unprotected sexual intercourse [37,38]. There is a paucity of research that applies syndemic theory to population burdens beyond HIV/AIDS [38].

We conceptualize IPV, depression, and inadequate nutrition as clustered and interacting epidemics that adversely affect mothers living in socio-economically disadvantaged households (i.e., households in the poorest social strata), leading to intergenerational health disparities. Fig 1 depicts a syndemic model [39] of intergenerational mother–child health disparities, which can be adapted more widely in terms of LMICs and exposure and outcome types. This conceptual model guided our statistical analysis. In this analysis, the intergenerational maternal child syndemic model focuses on child diarrheal symptoms and the maternal exposures of IPV, depression, and inadequate dietary iron intake. Our analysis was limited to child diarrheal disease and the Nepal context in part due to the lack of nationally representative surveys that concurrently measure child health outcomes alongside maternal IPV, mental health, and nutrition.

Syndemic theory is well positioned to provide insights into the production and amelioration of intergenerational health disparities in LMICs, yet mothers and their children have not been widely studied as a population for syndemics research. Women of reproductive age with children are at greater risk of IPV, depression, and nutritional inadequacies. A syndemic view of maternal-child health in LMICs can yield new insights into how health disparities are transmitted intergenerationally and under what conditions they emerge.

Building on Tsai and Venkataramani's [40] analytic approach, we conducted a secondary analysis of data from Nepal to test a novel syndemic model of intergenerational transmission of health disparities. This model proposes a syndemic of IPV, depression, and inadequate dietary consumption of iron among mothers that synergistically interact to magnify the population health burden of diarrhea among children under five. There are three interrelated research objectives that correspond to each postulate of syndemic theory:

*Objective 1 (Assessing Clustering):* Investigate the extent to which IPV, depression, and inadequate intake of iron-rich foods co-occur among mothers. We hypothesize that a subset of mothers will experience simultaneous IPV, depression, and lack of dietary iron sources.

*Objective 2 (Assessing Interaction)*: Investigate the extent to which the probability of child diarrheal morbidity is greater if mothers are triply exposed to IPV, depression, and inadequate intake of iron-rich foods, above and beyond the probabilities of morbidity if mothers are singly or doubly exposed to either condition. We hypothesize that mothers' IPV, depression, and lack of dietary iron sources interact with one another on multiplicative and additive scales to magnify the probability of child diarrheal symptoms.

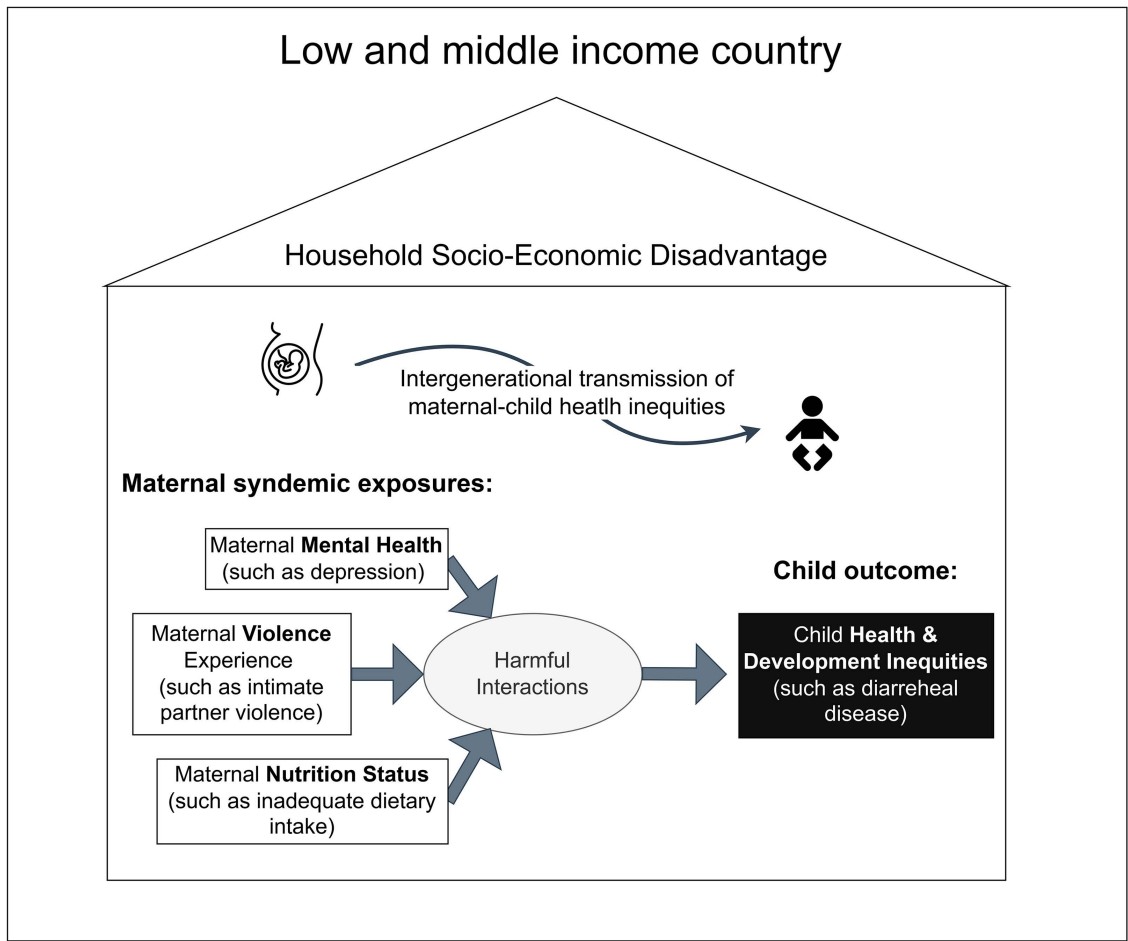

**Fig 1. Syndemic Model of Intergenerational Mother–Child Health Disparities.**

*Objective 3 (Assessing Social Factors and Contextual Forces)*: Investigate whether the degree of synergistic harm magnification between IPV, depression, inadequate dietary intake, and child diarrhea is greater among more disadvantaged households. We hypothesize that (1) the proportion of triply exposed mothers will be greater among households that are more (vs. less) disadvantaged, and (2) the three-term interaction involving maternal IPV, depression, and inadequate dietary intake will be statistically significant only among those in more disadvantaged households.

## Methods

### Research setting

Nepal is a post conflict, low-income country, bordering China and India, with a recently established democratic government. The nation transitioned from a monarchy to a federal democratic republic in 2008 following the 2006 Comprehensive Peace Agreement, marking the end of a decade-long civil conflict (1996–2006) [41]. The nation is home to 97 social groups that are classified into seven categories (Brahmin, Chhetri, Newari, Adivasi-Janajati, Dalits, other terai castes, and Muslims) which are grouped into "upper" or "lower" castes. The upper castes (Brahmin, Chhetri, and Newari) enjoy both social and economic advantage, especially in urban areas. The diversity in ethnicities, languages, and traditions presents the challenge of caste discrimination, gender inequality, and social stratification affecting access to education,

water and sanitation facilities, and healthcare [42]. The caste system influences social dynamics in complex ways. In Nepal and other South Asian countries, economic status and social status are distinct and yield differing effects on access to services. Social status inequalities result in reduced social capital and are rooted in history, religion, and ideology. For example, lower castes lack social capital and are more likely to lack access to water and sanitation and health care services [42]. Although greater earnings may reduce economic inequalities, the same is not true for caste inequalities which determine the concentration of services and social stigmatization [41,42].

## Data source

We conducted a secondary analysis of data from the 2022 Nepal Demographic and Health Survey (DHS), a nationally representative survey of households across Nepal's provinces and rural/urban areas, including noninstitutionalized women between the ages of 15–49 years. The 2022 Nepal DHS was used as the data source because it was the only publicly available dataset (in 2024) that concurrently measured all the necessary constructs (maternal IPV exposure, maternal depression, maternal nutrition, and child infectious disease symptoms) in an LMIC setting. The DHS has been implemented in Nepal since 1987. The 2022 Nepal DHS (implemented from January 5 to June 22, 2022) is the most recent survey implementation [43]. All women of reproductive age (15–49 years) in a selected household were surveyed about fertility, family planning, age of marriage, education, and child health. The response rate for women was 97% [44]. In a subset of randomly selected households, one woman of reproductive age (15–49 years) was randomly selected to complete the domestic violence module [45]. The questionnaires were programmed into tablets for computer-assisted personal interviewing in Nepali, Maithili, and Bhojpuri. The 2022 Nepal DHS followed a two-stage cluster sampling procedure [46]. We applied the DHS-provided sampling weights to generate nationally representative estimates.

## Measures

### Maternal syndemic exposures

We considered three maternal adversities (IPV, depression, and dietary iron intake) as syndemic exposures. Due to the cross-sectional DHS data source and differing self-reported recall periods we could not guarantee the correct temporal sequence between the maternal adversities. Each measure is detailed below.

**IPV (Previous year)** Women who reported ever having an intimate partner (husband or boyfriend) were asked questions about experiences of previous-year IPV using items of harmful behaviors that correspond to physical (seven items), emotional (three items), and sexual (three items) victimization. The behavioral measurement approach, which is similar to the Conflict and Tactics Scale [47], is preferred for IPV measurement as participants are provided multiple opportunities to report a range of harmful behaviors, thereby increasing sensitivity in detecting victimization [48–50]. A binary variable was created to denote exposure to any form of IPV (physical, emotional, or sexual) in the past year as exposed, otherwise responses were coded as unexposed (referent category).

*Depression symptoms (Previous 2 weeks)* Depressive symptoms were measured using the nine-item version of the Patient Health Questionnaire (PHQ-9) (sample α = 0.86). The PHQ-9 scores the nine DSM-IV depression criteria on a Likert-type scale. Participants are asked to report symptoms over a two-week period. The PHQ-9 was developed and tested for validity against using a psychiatric interview in a US-based clinical sample of primary care and obstetrics and gynecology patients [51]. A cut off score of at least 10 yielded optimal sensitivity (95%) and specificity (84%) and denotes moderate to severe depression [51]. The PHQ-9 has been adapted and validated for use in the Nepalese context [52]. Due to the sample size, a binary measure of depression could not be accommodated for the three-term interaction. For this reason, we report the prevalence of moderate/severe depression (i.e., as a binary measure) in the descriptive statistics but operationalized depression symptom severity as a numerical score count in the regression models.

*Iron-rich food intake (Previous day)* Women's dietary intake, a measure of short-term diet quality, was assessed using The Minimum Dietary Diversity for Women (MDD-W) tool. This tool captures population-level dietary patterns and

has been validated for women aged 15–49 years [53]. Using list recall, the tool measures intake of various food items during the past 24 hours. Foods rich in iron include organ meats, red flesh mammal's meats, poultry and other white meats, fish and seafood, and dark green leafy vegetables [53]. It was deemed important to include both plant and animal sources of iron since about 7% of people in Nepal are vegetarian [54] and vegetarianism is related to one's caste [55]. Caste in turn is related to socioeconomic status and household water and sanitation infrastructure [42]. Thus, to reduce biased estimates relative to the outcome of child diarrhea, we did not make iron consumption contingent on not being a vegetarian (separate sensitive analyses were run to assess animal source protein consumption). We operationalized the consumption of iron rich foods as binary: women who reported no dietary iron intake in the past day and women who consumed at least one plant or animal iron-rich food source (referent).

**Population health burden outcome:** Child Diarrhea Morbidity (Within the Previous 2 Weeks): One single survey question was used to create a binary outcome denoting the presence or absence of diarrhea. Mothers with biological children under the age of 5 years living in the same household were asked to report whether each child "had diarrhea in the last two weeks." All responses in the affirmative were coded as the child having diarrhea morbidities that are indicative of acute diarrhea.

**Contextual variable (Household wealth):** The household wealth index was constructed by the DHS using principal component analysis applied to household survey questions concerning ownership of consumer products and residence. Households are placed on a standardized scale of relative wealth and divided into ordinal wealth quintiles: poorest, poor, middle, less poor, least poor. Due to sample size limitations, we categorized households as 'most disadvantaged' if they were in the bottom two quintiles and 'less disadvantaged' if they were in the top three quintiles. Though we would have liked to assess more nuanced stratification patterns by quintile of household wealth, the three term interaction regression models required a larger sample size per wealth strata than was possible with the Nepal 2022 DHS.

## Covariates

The analysis adjusted for the set of variables that were thought to potentially confound the relationship between each maternal exposure and the outcome. Adjustment for all potential confounders is necessary because syndemic theory is interested in whether one can intervene on any syndemic exposure to disrupt synergistic harms [56,57] and the policy and program implementation implications of the interaction analysis are substantively different from other ways that the theory of syndemics has been operationalized [39]. Thus, confounding effects for all maternal exposures and the child diarrhea outcome must be mitigated to assess the total effect of the syndemic co-occurrence/clustering on infectious disease vulnerability.

To identify an appropriate set of control covariates in this observational analysis, a directed acyclic graph (DAG) was constructed. Fig A in S1 Text illustrates the hypothesized data generating process. The lead author developed this DAG [58] based on the available literature base concerning maternal health and child diarrhea in the Nepalese context [6,42,59–65]. To estimate the total estimated effect, analyses controlled for potentially confounding variables informed by the DAG: Food insecurity, urban/rural residence, women's empowerment, women's education, caste membership, household water treatment, and water and sanitation facility access (handwashing and sanitation). Table A in S1 Text outlines the measurement and operationalization of each covariate.

## Analysis

All statistical analyses were conducted in Stata version 17 (StataCorp LLC, College Station, Tex.). The subpopulation of interest included female participants who were randomly selected for the domestic violence module and had at least one child under five years. Mother-child dyads represent the unit of analysis; live children (under five years old) living in the same household as their biological mothers were linked to create a dataset containing maternal and household

characteristics for each mother–child dyad. A participant flow diagram is presented in Fig 2: 3.6% of the analytical sample had missing covariate information. We conducted complete case analysis using listwise deletion.

All analyses accounted for the complex sampling strategy by applying the primary sampling unit, strata, and weight for women selected into the domestic violence module. Taylor series linearization was used to produce design-adjusted standard errors. Stata's "singleunit(certainty)" option was used, which causes strata with single sampling units to be treated as certainty units which do not contribute to the standard error.

Using the *svyset* command, a two-stage stratified sampling design with weights was declared for inferential statistics. Stage 1 represented the primary sampling unit and stage two represented each sampled household. The second stage was deemed necessary as some mothers had more than one child. The mother–child dyads were clustered at the household level. The second sampling stage allowed for the correct estimation of standard errors for cases where multiple children originated from the same biological mother.

## Ethics statement

Since we conducted a secondary data analysis of the existing 2022 Nepal Demographic and Health Survey, ethical clearance was not required for the present analysis. According to 2022 Nepal Demographic and Health survey documentation, the Nepal Health Research Council and the ICF Institutional Review Board reviewed the survey protocol and written consent from the household head was required to carry out the interviews [43]. Anonymized and de-identified data were first accessed August 20th of 2023. We did not have access to information that could identify individual participants.

## Analytical strategy for syndemic clustering by household wealth quintile

The first analytical step involved testing for clustering. The clustering of maternal IPV, depression, and inadequate dietary iron intake was assessed using descriptive statistics. The degree of clustering in each household wealth level was examined to investigate whether population proportions were more prevalent among the most disadvantaged compared to the least disadvantaged households.

## Analytical strategy for syndemic interactions

The second step was testing statistical interactions on additive and multiplicative scales. This involved fitting fully saturated regression models [40] with three-order product terms involving the joint effects of maternal IPV, depression, and dietary iron intake on child diarrhea. The regression model also included the set of covariates previously conceptualized as potential confounders.

To test for interactions on the multiplicative scale, a multivariable logistic regression model was used (Equation A in S1 Text). Interaction on the multiplicative scale assesses whether the effect of three exposures occurring together exceeds the product effects of the three exposures considered separately and all possible two-way interactions. Further, using the logistic regression model, predicted probabilities at representative values of the maternal syndemic exposures were computed [66]. Predicted probabilities of child diarrhea were computed for one unit increases in maternal depression in the following maternal groups: (1) IPV exposed and absent consumption of dietary iron, (2) IPV exposed and consumed dietary iron, (3) IPV unexposed and absent consumption of dietary iron, (4) IPV unexposed and consumed dietary iron.

A second multivariable linear probability model with robust standard errors was constructed to test for interaction on the additive scale (Equation B in S1 Text), as per the approach adopted by Chakrapani et al. [67]. Statistical significance of the three-way product term is interpreted as statistical interaction on the additive scale wherein the triple effect of IPV, depression, and no dietary iron intake is greater than the sum of each single term and second order interaction term.

A subpopulation analysis involving the two household-wealth groups was conducted for the multiplicative and additive scale regression models. The models represented by Equations A and B in S1 Text were stratified by the binary household

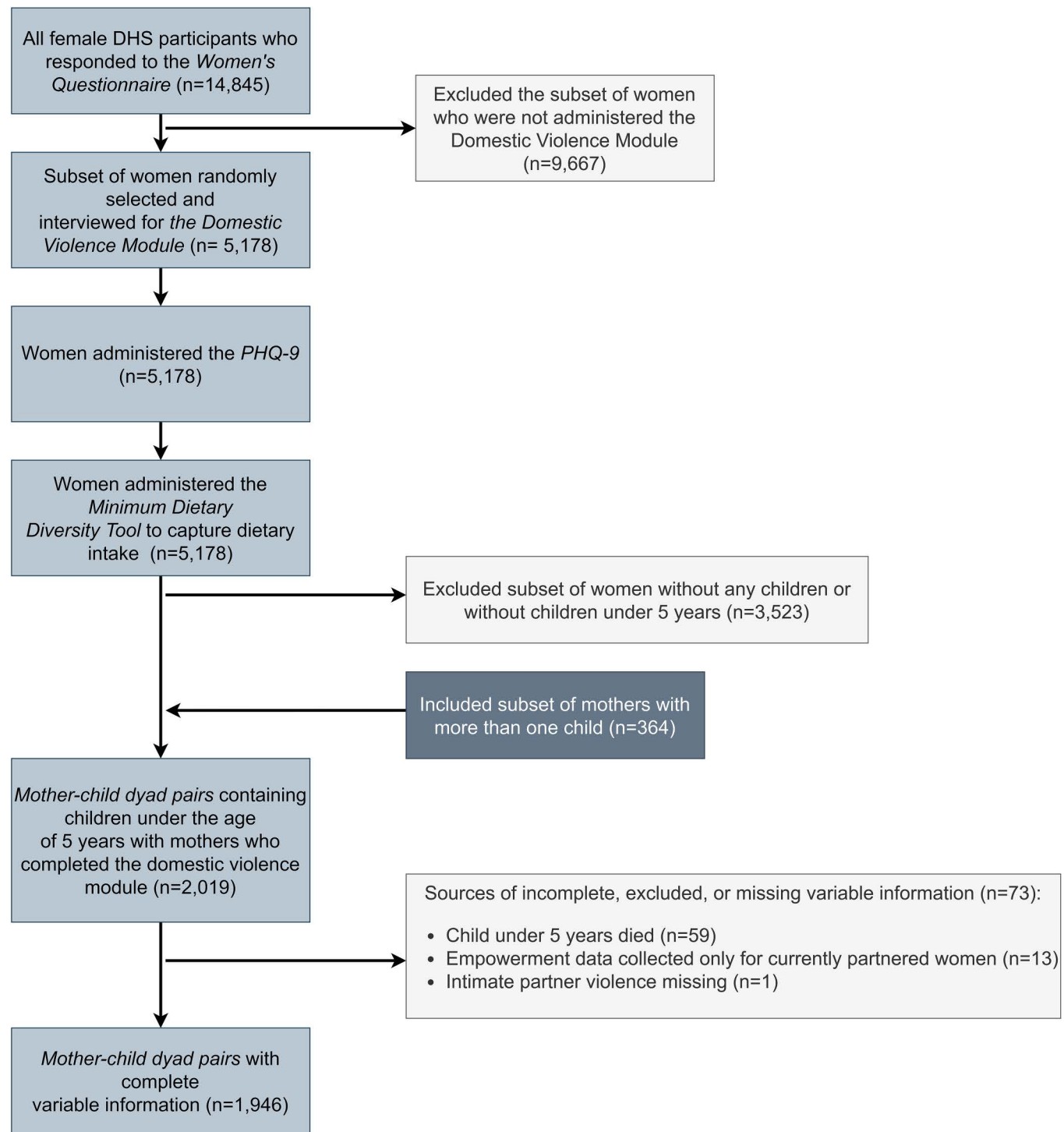

**Fig 2. Participant Flow Diagram, Culminating in the Final Analytical Sample.**

wealth contextual variable. The significance of the multiplicative (Equation A in S1 Text) and additive interactions (Equation B in S1 Text) were assessed in the most disadvantaged households (poorest and poor) and in more advantaged households (middle, less poor, least poor).

### Sensitivity analyses

Two sensitivity analyses were conducted to assess robustness of the main findings. First, calculations of the relative excess risk due to the three-term interaction (RERI) were calculated using guidance from Katsoulis et al. [68] and Vanderweele [57] as a sensitivity analysis to the linear probability model. This is presented in Table B S1 Text. Second, three binary variables for each form of previous year IPV (physical, sexual, and emotional) were created, wherein the referent category represented not being exposed to the specific form of violence. Iron-rich food intake was operationalized to include the intake of at least one heme source of iron (organ meats, red flesh mammal meats, poultry and other white meats, or fish and seafood). Considering the alternate operationalizations for IPV and iron-rich food intake, four additional interaction models were computed: (1) any previous year IPV and heme iron intake, (2) physical previous year IPV and heme iron intake, (3) emotional previous year IPV and heme iron intake, (4) sexual previous year IPV and heme iron intake. These models are presented in Tables C through G in S1 Text.

## Results

### Syndemic clustering and context

Descriptive statistics for clustering are presented in Table 1. The population prevalence of single exposure to IPV, moderate/severe depression, and inadequate dietary iron intake was greater among mothers in the poorest households, compared to mothers in the least poor households. There was less evidence that double and triple exposure to IPV, depression, and inadequate dietary iron intake was more prevalent among mothers from the poorest households compared to the least poor households. Only concurrent exposure to IPV and inadequate dietary iron intake was significantly greater among women in the poorest households (5.75%, 95% CI: 3.80%, 8.62%), compared to women in the least poor households (1.46%, 95% CI: 0.47%, 4.44%). The population prevalence of child diarrheal symptoms did not differ between the poorest and least poor households.

430 per 100,000 Nepalese mothers in the poorest households and 470 per 100,000 Nepalese mothers in the least poor households were triply exposed (IPV, severe depression, and absent dietary iron intake). The population prevalence of triply exposed mothers was rare (310 per 100,000 Nepalese mothers with children under five years across all wealth quintiles) and did not differ by household wealth quintile.

### Syndemic interactions and context

Table 2 presents the joint effects for the three-term syndemic interaction model (using multiplicative and additive scales) and computed among the most disadvantaged households (poorer and poor levels of household wealth) and least disadvantaged households (middle, less poor, and least poor levels of household wealth). The central finding is that in disadvantaged households, the syndemic interaction significantly increases the odds of child diarrhea, but not in advantaged households.

### Interactions on the multiplicative scale

The multiplicative models are presented in Table 2. Only in more disadvantaged households (Model 1a, Table 2) was there evidence to support the presence of a multiplicative interaction on child diarrhea when mothers were triply exposed (IPV+Depression+Absent Iron).

In more disadvantaged households each unit increase in maternal depression among mothers exposed to IPV and not consuming dietary iron (compared to mothers who did not experience IPV and were consuming dietary iron) increased

**Table 1. Population proportion exposed to each syndemic condition by household wealth in Nepal.**

| Syndemic Variable | Poorest % (95% CI) | Poorer % (95% CI) | Middle % (95% CI) | Less Poor % (95% CI) | Least Poor % (95% CI) | Overall % (95% CI) |
|---|---|---|---|---|---|---|
| **Maternal syndemic exposures** | | | | | | |
| IPV | 19.69 (15.50, 24.69) | 27.21 (21.48, 33.81) | 22.70 (16.90, 29.78) | 15.09 (9.87, 22.38) | 10.22 (6.28, 16.20)* | 19.57 (17.23, 22.13) |
| Depression score | 2.82 (2.46, 3.17) | 2.67 (2.18, 3.15) | 2.28 (1.83, 2.72) | 2.90 (2.38, 3.42) | 2.01 (1.52, 2.51)* | 2.56 (2.33, 2.78) |
| Moderate/ Severe depression | 5.28 (3.65, 7.59) | 5.70 (3.34, 9.57) | 4.95 (3.06, 7.91) | 5.62 (2.99, 10.32) | 1.34 (0.48, 3.74)* | 4.74 (3.67, 6.12) |
| Absent dietary iron intake | 26.57 (21.59, 32.23) | 31.15 (25.42, 37.52) | 31.28 (24.71, 38.70) | 28.52 (21.98, 36.10) | 16.67 (11.44. 23.66)* | 27.35 (24.25, 30.69) |
| Depression +Absent iron | 1.24 (0.54, 2.82) | 0.80 (0.26, 2.44) | 1.75 (0.82, 3.69) | 2.02 (0.88, 4.56) | 0.47 (0.07, 3.30) | 1.28 (0.79, 2.09) |
| IPV+Depression | 2.60 (1.58, 4.26) | 3.49 (1.78, 6.74) | 3.38 (1.86, 6.06) | 1.90 (0.77, 4.60) | 1.34 (0.48, 3.74) | 2.63 (1.86, 3.70) |
| IPV+Absent iron | 5.75 (3.80, 8.62) | 10.89 (7.26, 16.02) | 8.31 (5.03, 13.43) | 5.47 (2.58, 11.22) | 1.46 (0.47, 4.44)* | 6.64 (5.17, 8.50, |
| IPV+ Depression+ Absent iron | 0.43 (0.16, 1.11) | 0.80 (0.26, 2.44) | 1.44 (0.63, 3.25) | 0.94 (0.29, 2.97) | 0.47 (0.07, 3.30) | 0.83 (0.43, 1.56) |
| **Child syndemic outcome** | | | | | | |
| Diarrheal symptoms | 8.86 (7.35, 10.65) | 11.38 (9.13, 14.10) | 12.65 (10.13, 15.69) | 11.09 (8.97, 13.62) | 7.61 (5.53, 10.38) | 10.40 (9.16, 11.80) |
| **Mother–child syndemic** | | | | | | |
| IPV+Depres-sion+ Absent iron+Child Diarrhea | 0.21 (0.05, 0.84) | | 0.40 (0.12, 1.37) | | | 0.31 (0.12, 0.81) |

The total number of unique women who had at least one biological child under the age of 5 years and responded to the domestic violence module was 1,655 and was the population used to calculate the prevalence of the maternal syndemic exposures. The total number of mother–child dyad pairs (n = 2,019) was the population used to calculate the prevalence of the mother–child syndemic. The proportion of diarrheal symptoms was calculated using the total number of children <5 years n = 5,372.

The PHQ-9 was used to measure the depression score; to calculate the prevalence of each combination of syndemic variables, a PHQ-9 cut off score of at least 10 was used and corresponded to moderate/severe depressive symptoms.

'*' denotes statistically significant difference between the poorest and least poor household wealth groups, using the two tailed Wald test at an alpha of 0.05.

The descriptive statistics account for the primary sampling units, stratification, and weights.

Due to low cell sizes, the prevalence of the mother–child syndemic could not be computed by household wealth quintile and was instead computed for (1) poorest and least poor and (2) middle, less poor, and least poor.

the odds of a child under the age of 5 years having diarrhea by 32.1% (aOR: 1.321, 95% CI: 1.000, 1.745, $P$=0.05). This three-term interaction estimate is above and beyond the isolated and/or joint effects of IPV and dietary iron consumption. Among less disadvantaged households (Model 2a, Table 2), the joint effect of IPV, depression, and absent dietary iron was not statistically significant (aOR: 1.090, 95% 95: 0.789, 1.507, $P$=0.599). However, mothers' isolated exposures to IPV and depression were significantly and positively related to child diarrhea.

### Interactions on the additive scale

A similar pattern of results was noted when using the linear probability model, although $P$ values were marginally insignificant in the more disadvantaged households. Only among mothers living in more disadvantaged households (Table 2,

**Table 2. Joint effects of maternal syndemic interactions and child diarrhea (2022 Nepal DHS).**

| Maternal syndemic exposures | Model 1a: Multiplicative Scale Most Disadvantaged households<br>aOR<br>P value<br>95% CI | Model 1b: Additive Scale Most Disadvantaged households<br>B<br>P value<br>95% CI | Model 2a: Multiplicative Scale Less Disadvantaged<br>aOR<br>P value<br>95% CI | Model 2b: Additive Scale Less Disadvantaged<br>B<br>P value<br>95% CI |
|---|---|---|---|---|
| Mother experienced any IPV in the past 12 months | | | | |
| + IPV (ref: unexposed to IPV) | 0.536 | -0.038 | 3.842* | 0.152 |
| | 0.184 | 0.160 | 0.033 | 0.098 |
| | [0.214, 1.345] | [-0.091, 0.015] | [1.114, 13.257] | [-0.028, 0.331] |
| Mother's depression score (one unit increase) | 1.073 | 0.007 | 1.113* | 0.011 |
| | 0.09 | 0.090 | 0.045 | 0.078 |
| | [0.989, 1.165] | [-0.001, 0.016] | [1.002, 1.236] | [-0.001, 0.022] |
| Mother's iron rich food consumption | | | | |
| - Iron (ref: consumed iron rich foods) | 1.109 | 0.006 | 0.679 | -0.018 |
| | 0.789 | 0.811 | 0.391 | 0.476 |
| | [0.518, 2.375] | [-0.046, 0.059] | [0.279, 1.649] | [-0.067, 0.031] |
| Mother exposed to any IPV in the past 12 months + Depression score | | | | |
| | 0.971 | -0.005 | 0.833 | -0.020 |
| | 0.623 | 0.361 | 0.108 | 0.117 |
| | [0.865, 1.091] | [-0.014, 0.005] | [0.666, 1.041] | [-0.045, 0.005] |
| Mother did not consume iron rich foods + Depression score | | | | |
| | 0.887 | -0.011 | 1.017 | -0.003 |
| | 0.263 | 0.211 | 0.869 | 0.721 |
| | [0.718, 1.095] | [-0.027, 0.006] | [0.829, 1.248] | [-0.021, 0.014] |
| Mother did not consume iron rich foods + Mother exposed to any IPV in past 12 months | | | | |
| | 0.502 | -0.057 | 1.121 | 0.002 |
| | 0.477 | 0.303 | 0.917 | 0.990 |
| | [0.075, 3.365] | [-0.167, 0.052] | [0.129, 9.743] | [-0.335, 0.340] |
| Mother exposed to any IPV in the past 12 months + Mother did not consume iron rich foods + Depression score | | | | |
| | 1.321* | 0.027 | 1.090 | 0.014 |
| | 0.050 | 0.067 | 0.599 | 0.499 |
| | [1.000, 1.745] | [-0.002, 0.055] | [0.789, 1.507] | [-0.027, 0.056] |
| Number of observations in each subpopulation | 1,139 | 1,139 | 807 | 807 |
| Omnibus model fit test | F(16, 447)=2.06, P=0.009 | F(16, 447)=1.46, P=0.1087 | F(16, 447)=1.80, P=0.029 | F(16, 447)=1.35, P=0.1623 |

* $P < .05$. All models controlled for the same set of covariates deemed to be confounders based on the DAG presented in Fig A in S1 Text: Food insecurity, urban/rural residence, woman's empowerment, woman's education, caste membership, household water treatment, and water and sanitation facility access (handwashing and sanitation).

Model 1b) was the three-term interaction (IPV + Depression + Absent Iron) associated with an increase in child diarrhea (B = 0.027, 95% CI: -0.002, 0.055, $P = 0.067$). This association suggests that for triply exposed mothers, the prevalence of child diarrhea is greater than would be expected based on the sum of each double exposure and each single exposure.

Among mothers in less disadvantaged households (Table 2, Model 2b), the three-term interaction was nonsignificant (B=0.014, 95% CI:-0.027, 0.056 P=0.499), indicating the lack of synergy on the additive scale.

**Predicted probabilities**

The pattern of syndemic interactions was visualized using model predicted probabilities: marginal effects at representative values of the syndemic exposures, holding all covariates at their mean levels. Fig 3 displays the predicted probabilities stemming from the logistic regression models: Model 1a (more disadvantaged households) and Model 2a (less disadvantaged households). The adjusted odds ratio and 95% CI for the relationship between maternal depression and child diarrhea among each unique combination of maternal syndemic exposures involving IPV and dietary iron consumption is also presented in Fig 3.

The synergistic interaction (IPV+Depression+Absent Iron) that is associated with disadvantaged households is evident when comparing the blue line of plotted predicted probabilities between the top panel and the bottom panel in Fig 3. For mothers living in more disadvantaged households who are exposed to IPV and not consuming adequate dietary iron, increasing depression increases the risk of child diarrhea (aOR: 1.22, 95% CI: 1.04, 1.44, P=0.02). However, the same is not true for mothers in less disadvantaged households where depression symptoms were unrelated to child diarrhea among mothers who were exposed to IPV and not consuming dietary iron sources (aOR: 1.03, 95% CI: 0.87, 1.22, P=0.75).

For mothers in more disadvantaged households, when maternal depression is less than 10 on the PHQ-9, the predicted probability of child diarrhea does not diverge according to IPV exposure or dietary iron intake. However, as mothers experience moderate and severe depression (PHQ-9 score ≥ 10), the predicted probability of child diarrhea is magnified among mothers exposed to IPV and who do not consume dietary sources of iron.

Marginal effects at representative values are an additional tool help to further interpret the additive syndemic interaction. Note that the marginal effects at representative values are not displayed in any table but are derived from Fig 2. In more disadvantaged households, among Nepalese mothers who do not display syndemic vulnerability (i.e., no depressive symptoms (PHQ-9=0), unexposed to previous year IPV, and are consuming at least one source of dietary iron) the probability that their children (under the age of five years) have diarrhea is 7.5% 95%CI: 4.7%, 10.3%, P<0.001). However, in the same households, for Nepalese mothers who display syndemic vulnerability (severe depressive symptoms (PHQ-9=20), exposed to previous year IPV, and not consuming dietary iron sources), the probability that their children have diarrhea rises to 53.5% (95%CI: 1.3% to 100% P=0.045). For mothers in advantaged households, the probability that a child under 5 has diarrhea is 8.1% (95%CI: 4.3%, 11.8%, P=0.00) when mothers display no syndemic vulnerability. For mothers exposed to IPV, with severe depression [PHQ-9=20], and inadequate dietary iron intake, the probability that children (under five) have diarrhea is nonsignificant (29.8%, 95%CI: -15.4%, 75.0% P=0.195) in advantaged households. While predicted probabilities help to better understand the interaction patterns, in our analysis they at times imprecise due to the limited number of mothers concurrently exposed to all syndemic conditions.

**Sensitivity analyses**

The sensitivity analyses reflected the main analysis. For the RERI, there was evidence of additive interactions for a 10 unit increase in depression score (Table B in S1 Text), despite the limitations in applying the RERI to nonrare outcomes when using logistic models. Considering the intake of heme iron sources and measuring depression using the PHQ-9, the three-term joint effects interaction on the multiplicative scale was significant for any IPV, physical IPV, and emotional IPV. The estimate for sexual IPV did not reach the threshold of significance using the multiplicative scale. On the additive scale, the three-term interaction was only significant for physical IPV and marginally insignificant for any IPV. Thus, three-term interactions involving dietary heme iron intake, depression score, and any or physical IPV were in the direction anticipated and significant on additive and multiplicative scales (Tables C through G in S1 Text)

## Subpopulation of Disadvantaged Households (Model 1)

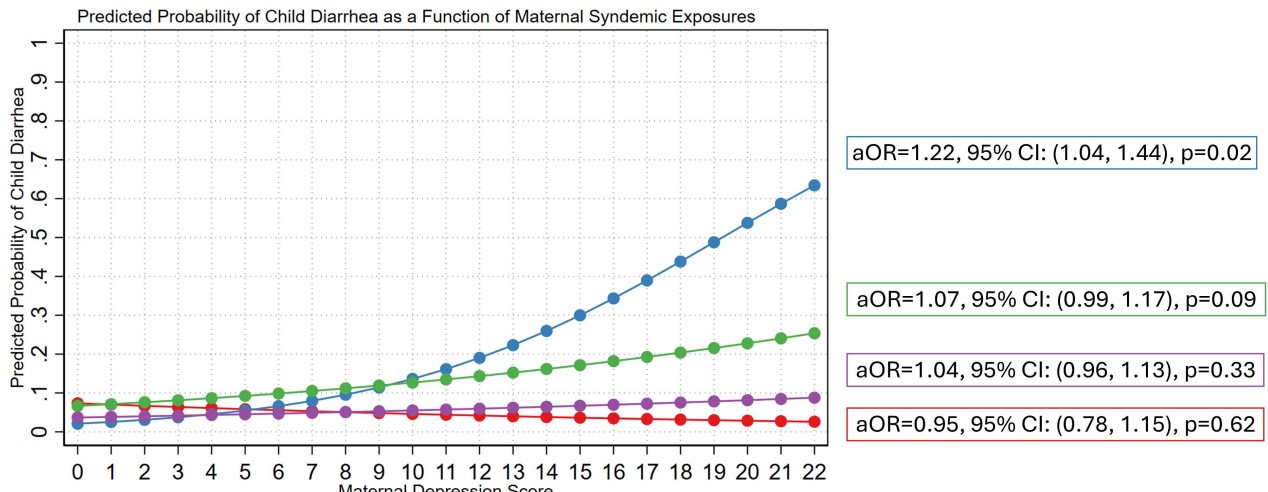

## Subpopulation of Advantaged Households (Model 2)

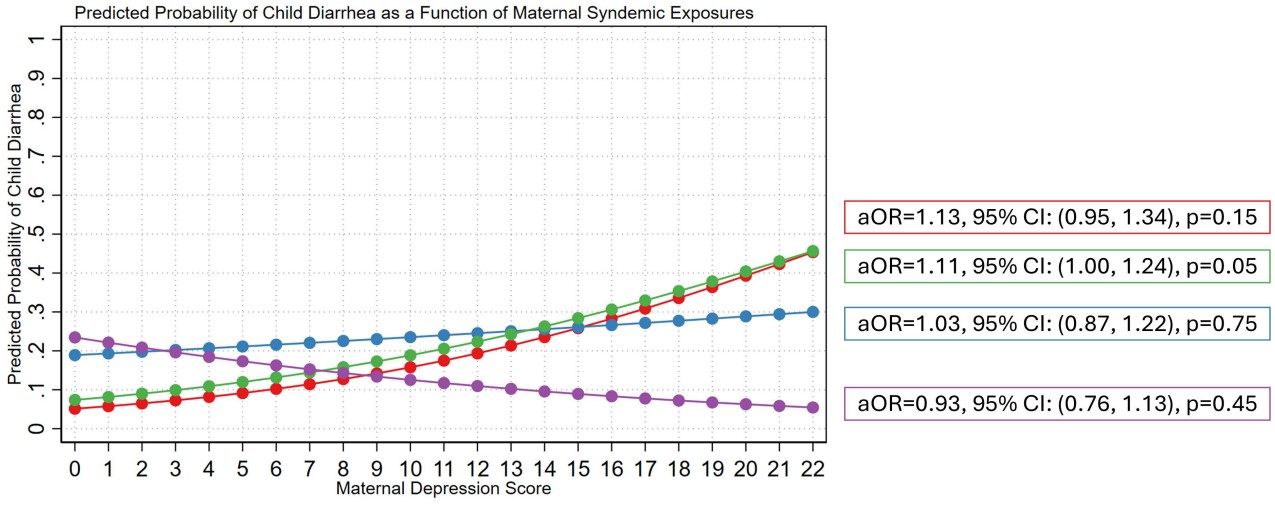

## Legend

| | | | |
|---|---|---|---|
| —●— | - Iron, - IPV | —●— | - Iron, + IPV |
| —●— | + Iron, - IPV | —●— | + Iron, + IPV |

**Fig 3. Model predicted probabilities that visually display hypothesized syndemic interactions.** Note: The predicted probabilities were computed for model 1a and model 2a displayed in Table 2. All covariates were held constant at their mean levels.

## Discussion

This secondary analysis of the 2022 Nepal DHS sought to test a novel syndemic model of intergenerational mother–child health disparities in LMIC. Drawing on the existing postulates of syndemic theory, [31] this analysis tested the degree to which mothers' simultaneous exposure to IPV in the past year, depression in the past two weeks, and absent dietary iron consumption in the past day had a synergistic relationship with diarrheal symptoms in the past 2 weeks among children under five years. The role of the household socioeconomic environment in influencing the clustering and interaction between maternal IPV, depression, and dietary iron consumption was also assessed. Our findings provide emerging evidence for synergistic maternal interactions on child diarrheal disease in more disadvantaged households.

Three key findings emerged from our analysis. First, regarding clustering there was no evidence that mothers from more disadvantaged households faced a greater triple burden (IPV + Depression + Absent Iron). This finding may reflect limitations in power (further discussed in the limitations section). However, compared to the least disadvantaged households, mothers from the most disadvantaged households faced greater single exposure to any IPV, moderate/severe depression, and inadequate dietary iron intake. Second, the synergistic interaction between maternal IPV, depression, and inadequate dietary iron intake on child diarrhea (using multiplicative and additive scales) emerged only in more disadvantaged households. Third, the synergistic interaction association with child diarrhea emerged when mothers reached levels of moderate and severe depression.

Through other scholarship has conceptually positioned syndemics as a useful framework for child health and development inequities such as diarrheal disease [38,69–71], empirical scholarship is only yet emerging. For example, an empirical analysis focused on the South African context reported synergistic additive and multiplicative interactions on ambulatory child diarrheal illness when mothers were exposed to IPV and HIV [72]. Other additive and multiplicative interactions on child motor delays were noted for household food insecurity and maternal HIV. Considering this South African study, our findings further support evidence of synergistic harm to child health and development when mothers are exposed to multiple adversities, including IPV [72]. Additional research is needed to reduce the empirical gap in applying syndemic theory to maternal child health disparities [38]. Future research can extend the theoretical model in Fig 1 across LMICs settings and different profiles of child health and development inequities as well as maternal adversities related to violence, depression, and nutrition.

### Public health approaches to addressing maternal-child syndemics

Our results indicate that the intergenerational mother–child syndemic affects a small proportion of Nepalese mothers among disadvantaged households but potentially yields a synergistic magnification in the probability of child diarrhea. To mitigate the intergenerational health disparity, a proportionate universal public health approach is recommended, [73] whereby public health action to prevent the intergenerational syndemic is universal but tailored to a specific scale and intensity based on maternal indicators [73]. For example, all families expecting newborns each year could be contacted by a community health worker or public health nurse to assess for maternal syndemic conditions. In this prenatal period, families living in the most disadvantaged households who screen positive or are at risk for triple adversities (IPV + Depression + Absent Iron Consumption) could be matched with additional in-home/community supports providing continuous support in the prenatal and postnatal period. These supports could be aimed at disrupting one of the three harmful maternal exposures. Families who do not screen positive or reside in the least disadvantaged households will be offered more general support that is less frequent and of lower intensity. To address the sociopolitical environment, national and regional governments can in parallel raise the standard of living among families living in the most disadvantaged households who are expecting newborns.

As other research has demonstrated, Nepal has made exemplary progress in reducing chronic malnutrition and under-five child mortality [59]. Despite this progress, certain regions in Nepal (the sub provincial districts of Achham and

Rasuwa) did not experience reduction in under-five child diarrhea [74]. In these regions, our findings indicate two potential intervention strategies. First, there was evidence of synergistic magnification in child diarrhea if mothers from disadvantaged households were exposed to IPV, depression, and absent iron intake. To disrupt harmful synergies, potential intervention pathways can involve the i) prevention and mitigation of IPV harms, such as the treatment of depression, ii) maternal nutrition support, and iii) reducing structural inequities related to poverty and marginalization.

According to syndemic theory and under the condition of causal interaction, intervening to eliminate any one harmful exposure may reduce the probability of child diarrheal disease to a greater extent than would be predicted were no synergies apparent. Given the increased prevalence of maternal IPV in the most disadvantaged households, acting to prevent IPV and addressing its sequelae on mental health and dietary patterns could be a strategic programmatic target. Alternatively, maternal depression may also be a strategic entry point into marital conflict and IPV. Second, improving household wealth may disrupt the harmful synergies. However, more research is needed to isolate specific aspects of household wealth that yield causal and downstream impacts on maternal wellbeing and ultimately child diarrhea.

In less disadvantaged households, there was no evidence of harmful synergies on the probability of child diarrhea if mothers were exposed to IPV, depression, and inadequate dietary iron consumption. However, singular exposure to maternal IPV or increasing depression were associated with greater child diarrhea. This indicates that both IPV and depression should be targeted to address their independent effects on child diarrhea. However, intervening on one maternal health adversity may not impact the harmful synergistic relationship involving the other.

## Nepal Context: Strengthening trauma informed parenting through the existing safe motherhood program

Parents are the primary vehicle through which children receive nourishment and protection from enteric disease. For example, mothers play a direct role in child nutrition during infancy via breastfeeding. Mothers who bear parenting responsibilities while also exposed to violence and malnourished may be less likely or able to engage in parenting practices that are protective against child diarrhea and likely other child development outcomes. While living in an economically stressed household with an abusive and less supportive partner, mothers bearing the triple syndemic burden vulnerability (IPV + Depression + Absent Iron) may have less knowledge, ability, or agency to access services that can disrupt harmful interactions.

The behavioral parenting pathway within a context of household disadvantage could inform the integration of violence protections within maternal child health programs. For example, using a community health worker or public health nurse model, existing mother–child programs could be coupled with a risk of violence assessment for expecting mothers. Women deemed to be at risk of or exposed to IPV can be referred into a gender-transformative family parenting program (along with their spouses/partners and other in-laws) that better links families to health and social services and offers socioeconomic supports. Such an approach is sensitive to Nepal's gender norm context which may not be receptive to IPV-centered programs [63]. Established in 1997, Support to Safer Motherhood Program is focused on reducing financial barriers to accessing maternal and newborn health services, financial incentives covering transport fees for hospital-based delivery, and institutional delivery with skilled birth attendants. [75] However, the Support to Safer Motherhood Program is less focused on the postnatal care period and mainstreaming violence protections: two potential area of expansion and integration.

## Strengths

Regarding our conceptual contributions to the literature on syndemics, the present analysis improves the empirical testing of syndemic theory by (1) expanding on the existing methodological guidance for interaction testing [40] and (2) developing a new intergenerational mother–child syndemic model for LMICs. Assessing the degree to which context affects the clustering and interaction of syndemic exposures is a postulate of syndemic theory that has not been adequately

                                                                                        

PLOS Global Public Health

assessed in quantitative literature, which is more focused on interaction testing (to the extent that interactions are assessed at all). As demonstrated in the present analysis, conducting a subpopulation analysis using a fully saturated interaction model can additionally test under what contextual conditions syndemic interactions occur. Such an approach helps to identify counter-syndemics: a context or population where the syndemic does not occur [76]. Use of DAGs to illustrate the causal pathways and confounders also increases transparency [77]. Future studies can apply the intergenerational syndemic model involving gender-based violence as a maternal syndemic exposure to other leading causes of child morbidity and mortality in LMICs.

Further, using nationally representative data (such as the DHS) provides an opportunity for syndemic researchers to assess the public health significance of syndemics. Empirical investigations of syndemics could not solely focus on isolating interaction effects with *P* values of <0.05. Rather, examining the population burden of syndemic exposures can identify how many people are currently experiencing or at risk of experiencing the syndemic and can inform public health planning. In our analysis, the syndemic was associated with a greatly elevated burden of childhood diarrheal illness, but the population prevalence of mothers who were triply exposed was relatively low.

In terms of future research, additional qualitative investigation is needed to holistically investigate syndemic mechanisms, particularly involving context. Evidence of syndemic interactions is important but provides an incomplete understanding of why these syndemic conditions occur. To better inform targeted interventions, qualitative research in LMICs conducted with mothers experiencing syndemic vulnerability, their spouses, and service providers can probe local perceptions regarding why some mothers experience the triple burden of IPV, depression, and lack of dietary iron intake and how these conditions interact. Additionally, the role of fathers in the household and parenting environment, can be better integrated.

## Limitations

This study should be interpreted with several limitations in mind. First, because the DHS data are cross-sectional, we cannot establish cause-and-effect relationships. The timing of exposures and outcomes cannot be confirmed, so our findings should be understood as associations rather than causal effects.

Second, we examined several health indicators at the same time and calculated their overall population proportions. As a result, the sample size may not have been large enough to reliably estimate prevalence or differences between wealth groups. Stall and colleagues [78] note power issues, which may explain why the combined exposure to maternal health burdens (IPV, depression, and inadequate iron intake) did not differ significantly across wealth quintiles.

Third, this study is constrained by the measurement tools used in the DHS. The PHQ-9 may have been difficult for some participants to fully understand or relate to, which could have led to lower reported depression scores. In addition, the list-based recall method used to measure dietary diversity is only an indirect indicator of iron intake [53]. Compared with open-recall methods, list-based recall tends to lead to overreporting [79]. Dark leafy green vegetables, in particular, are often misreported, which may have resulted in an overestimation of iron-rich food consumption in our sample [79].

Child diarrhea was also measured using a single self-reported question. No visual aids or prospective diaries were used, and the survey did not capture important details such as how long symptoms lasted or whether blood was present. This means we could not assess prolonged, persistent, or chronic diarrhea. Moreover, research shows that longer recall periods lead to substantially lower reports of diarrhea among children in LMICs [80]. The DHS uses a two-week recall period, which increases the likelihood of underreporting [80]. As a result, the measurement of diarrhea in this study may contain considerable noise.

Due to sample size limitations it was not feasible to stratify our interaction models by caste. For this reason, we held caste constant in our analysis. The caste system in Nepal shapes social experiences and access to services in complex ways. Economic status and social status do not align neatly, and caste-based inequalities often have deeper and

more persistent effects than household wealth alone [42]. Lower-caste groups may have limited social capital and poorer access to water, sanitation, and health care, even when their household wealth is similar to that of higher-caste groups. While we were unable to analyze interactions by caste due to small subgroup sizes, caste remains an important contextual factor in syndemic processes in LMICs and should be explored in future research.

Finally, due to several methodological constraints (non-temporally aligned recall periods between the exposures and the outcome, the low prevalence of simultaneous exposure to all three maternal adversities, and dependence on a single maternal report of child diarrhea) causal interpretation of our results warrants caution. Together, these constraints likely reduced precision in the three-way interaction analyses, resulting in wide confidence intervals for certain predicted probabilities. Accordingly, the present estimates should be interpreted as indicative of potential interaction patterns rather than as precise measures of risk.

## Conclusion

This research rested a novel syndemic model of intergenerational mother–child health disparities in LMICs. Using data from the 2022 Nepal DHS and a robust quantitative analytical strategy for testing interactions by household wealth subpopulations, we provide evidence to support a mechanism whereby household disadvantage is associated with harmful interactions between the maternal health adversities of IPV, depression, and inadequate dietary iron consumptions, which in turn magnify the probability of diarrhea for under-five children above and beyond the individual effects of IPV, depression, and dietary iron deficiency. Though single exposure to maternal health conditions was more prevalent among the most disadvantaged households, little evidence supported a greater degree of clustering between the maternal health disparities among the disadvantaged households, compared to the advantaged households.

## Supporting information

**S1 Text. Supplementary Appendices Table of Contents.**
(DOCX)

## Author contributions

**Conceptualization:** Luissa Vahedi, Lindsay Stark.

**Formal analysis:** Luissa Vahedi.

**Investigation:** Luissa Vahedi.

**Methodology:** Luissa Vahedi, Ilana Seff, Alexander C. Tsai, Lora Iannotti.

**Visualization:** Luissa Vahedi.

**Writing – original draft:** Luissa Vahedi.

**Writing – review & editing:** Luissa Vahedi, Ilana Seff, Alexander C. Tsai, Lora Iannotti, Lindsay Stark.

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
