## [Decision Letter · Decision Letter 0]

3 Oct 2025

PGPH-D-25-01038

Parenting Under the Triple Burden of Violence, Depression, and Malnutrition: An Intergenerational Mother–Child Syndemic in Nepal

Dear Dr. Vahedi,

Thank you for submitting your manuscript to PLOS Global Public Health. After careful consideration, we feel that it has merit but does not fully meet PLOS Global Public Health’s publication criteria as it currently stands. Therefore, we invite you to submit a revised version of the manuscript that addresses the points raised during the review process.

Editor comments:

The reviewers and I found the manuscript to be well-written, and to offer a novel application of the syndemics framework to explore complex relationships between child and maternal health in Nepal. The methodology is meticulously described, which is also helpful. Please note that reviewers uploaded their detailed feedback as separate attachments, and please make sure to address those comments in the revisions.Both the reviewers and I found the introduction and the model presented to be a little difficult to follow. Many different topics and variables are presented, with varying degrees of explanation about how they relate to the model(s) to be tested. Streamlining this to focus more clearly on what is presented in Figure 1 and what will be tested would be helpful.The methods could also be streamlined, with technical details moved to an appendix, which would make it easier for the reader to understand the overall approach. Reviewers also had several questions about variables that need further explanation. The results and discussion would also benefit from clearer organization and streamlining as recommended by both reviewers.

We look forward to receiving your revised manuscript.

Kind regards,

Marie A. Brault, PhD

Academic Editor

Journal Requirements:

2. Your current Financial Disclosure states, “The author(s) received no specific funding for this work. ACT reports funding from U.S. National Institutes of Health K24DA061696-01. LV's doctoral studies were supported by The Social Sciences and Humanities Research Council of Canada (Doctoral Fellowship) and the P.E.O Scholar's Award.”. However, you did not provide a funding information in your online submission form. Please indicate by return email the full and correct funding information for your study and confirm the order in which funding contributions should appear. Please be sure to indicate whether the funders played any role in the study design, data collection and analysis, decision to publish, or preparation of the manuscript.

3. Please send a completed 'Competing Interests' statement, including any COIs declared by your co-authors. If you have no competing interests to declare, please state "The authors have declared that no competing interests exist". Otherwise please declare all competing interests beginning with the statement "I have read the journal's policy and the authors of this manuscript have the following competing interests:"

Additional Editor Comments (if provided):

Reviewers' comments:

Reviewer's Responses to Questions

**Comments to the Author**

1. Does this manuscript meet PLOS Global Public Health’s publication criteria?

Reviewer #1: Yes

Reviewer #2: Yes

2. Has the statistical analysis been performed appropriately and rigorously?

Reviewer #1: Yes

Reviewer #2: Yes

3. Have the authors made all data underlying the findings in their manuscript fully available (please refer to the Data Availability Statement at the start of the manuscript PDF file)?

Reviewer #1: Yes

Reviewer #2: Yes

4. Is the manuscript presented in an intelligible fashion and written in standard English?

Reviewer #1: Yes

Reviewer #2: Yes

Reviewer #1: Please see attached comments for further details. This study focuses on identifying the relationship between concurrent exposure of intimate partner violence, depression, and inadequate iron intake among mothers on diarrheal disease among children. The study used a national representative sample of women of reproductive age in Nepal. The findings suggest risk of diarrhea is higher in households with concurrent exposure, and that this is further exacerbated by living in a disadvantaged household.

Reviewer #2: Thank you for the opportunity to review this manuscript. Overall, the study is methodologically rigorous, ethically conducted, and addresses an important public health question that is highly relevant to global audiences. The research design is sound, the statistical analyses are appropriate, and the conclusions are well-supported by the data presented.

The manuscript is written in clear, standard English and is presented in an accessible format that will be understandable to a broad readership. The authors have also complied with the PLOS data availability requirements, ensuring transparency and reproducibility.

I commend the authors for their attention to detail in both the analysis and discussion. The study makes a valuable contribution to the field of global public health.

**Do you want your identity to be public for this peer review?** For information about this choice, including consent withdrawal, please see our Privacy Policy

Reviewer #1: No

Reviewer #2: **Yes:**  Nanki Singh

---

## [Decision Letter · Decision Letter 1]

13 Jan 2026

PGPH-D-25-01038R1

Parenting Under the Triple Burden of Violence, Depression, and Malnutrition: An Intergenerational Mother–Child Syndemic in Nepal

Dear Dr. Vahedi,

Thank you for submitting your manuscript to PLOS Global Public Health. After careful consideration, we feel that it has merit but does not fully meet PLOS Global Public Health’s publication criteria as it currently stands. Therefore, we invite you to submit a revised version of the manuscript that addresses the points raised during the review process.

Editor comments:

The reviewers and I appreciate the authors' responsiveness to the previous reviews and feel that the manuscript is much clearer.However, there are several areas where minor errors and inconsistencies need to be clarified (see below and attached for specific notes).

We look forward to receiving your revised manuscript.

Kind regards,

Marie A. Brault, PhD

Academic Editor

Journal Requirements:

Additional Editor Comments (if provided):

Reviewers' comments:

Reviewer's Responses to Questions

**Comments to the Author**

Reviewer #1: All comments have been addressed

Reviewer #2: All comments have been addressed

publication criteria?

Reviewer #1: Yes

Reviewer #2: Yes

3. Has the statistical analysis been performed appropriately and rigorously?

Reviewer #1: Yes

Reviewer #2: Yes

4. Have the authors made all data underlying the findings in their manuscript fully available (please refer to the Data Availability Statement at the start of the manuscript PDF file)?

Reviewer #1: Yes

Reviewer #2: Yes

5. Is the manuscript presented in an intelligible fashion and written in standard English?

Reviewer #1: Yes

Reviewer #2: Yes

Reviewer #1: See attached for very minor grammatical errors.

Reviewer #2: This study makes a strong contribution by advancing an intergenerational syndemic framework linking maternal intimate partner violence (IPV), depressive symptoms, and nutritional vulnerability to child diarrheal illness in Nepal. The syndemic framing is clear and theoretically grounded, with explicit objectives addressing clustering, interaction, and contextual forces. The analysis appropriately uses DHS data, accounting for survey design and the domestic violence module, and the interaction testing is carefully implemented using fully saturated models on both additive and multiplicative scales, supported by predicted probabilities. Limitations related to cross-sectional design, measurement, and statistical power are acknowledged transparently.

Several clarifications would strengthen alignment between claims and measures. Most notably, the nutrition exposure reflects reported consumption of iron-rich foods in the prior 24 hours instead of rather than malnutrition or iron deficiency, and should be framed as a short-term dietary quality indicator. In addition, the abstract appears to reverse additive and multiplicative interaction estimates (with the odds ratio and linear probability coefficient mislabeled), which should be corrected for consistency with Table 2. Minor typographical errors in the tables should also be addressed.

{ Your key interaction estimates are confusingly reported (OR vs B swapped in text)

In the abstract you report additive B=1.321 and multiplicative B=0.027, but in Table 2:

• Multiplicative: OR for 3-way term is 1.321

• Additive (LPM): B for 3-way term is 0.027

So the abstract has the scales/labels reversed.

Additional typos

“10.40 (9.16, 1180)” → likely “11.80”

“2.01 (1.52. 2.51)” → comma typo

“1.75 (0.0082, 3.69)” → likely “0.82” not “0.0082” }

Interpretation of the findings should remain cautious given discordant recall periods across exposures and outcomes, the rarity of concurrent exposure to all three maternal adversities, and reliance on a single maternal report of child diarrhea. These factors likely contribute to imprecision in three-way interaction estimates and wide confidence intervals for some predicted probabilities, which are best viewed as illustrative of interaction patterns rather than precise risk estimates. Despite these limitations, the consistency of results across modeling approaches and sensitivity analyses suggests that combined maternal vulnerabilities could meaningfully elevate child diarrhea risk in socioeconomically disadvantaged households.

Overall, this is a very strong study and it highlights the value of a syndemic framework for identifying contexts in which multiple maternal adversities interact to amplify child health inequities, while underscoring the need for longitudinal and mixed-methods research to clarify mechanisms and causal pathways.

**Do you want your identity to be public for this peer review?** For information about this choice, including consent withdrawal, please see our Privacy Policy

Reviewer #1: No

Reviewer #2: **Yes:**  Nanki Singh

---

## [Editor Report · Decision Letter 2]

28 Jan 2026

Parenting Under the Triple Burden of Violence, Depression, and Poor Diet Quality: An Intergenerational Mother–Child Syndemic in Nepal

PGPH-D-25-01038R2

Dear Ms. Vahedi,

We are pleased to inform you that your manuscript 'Parenting Under the Triple Burden of Violence, Depression, and Poor Diet Quality: An Intergenerational Mother–Child Syndemic in Nepal' has been provisionally accepted for publication in PLOS Global Public Health.

Best regards,

Marie A. Brault, PhD

Academic Editor